# Evaluation of the Mexican warning label nutrient profile on food products marketed in Mexico in 2016 and 2017: A cross-sectional analysis

Alejandra Contreras-Manzano[◉], Carlos Cruz-Casarrubias[◉], Ana Munguía[‡], Alejandra Jáuregui[◉*], Jorge Vargas-Meza[‡], Claudia Nieto[‡], Lizbeth Tolentino-Mayo[‡], Simón Barquera[◉]

Center for Research on Nutrition and Health, National Institute of Public Health, Cuernavaca, Morelos, Mexico

◉ These authors contributed equally to this work.
‡ These authors contributed equally to this work.
* alejandra.jauregui@insp.mx

**Data Availability Statement:** All relevant data are within the manuscript and its Supporting information files.

## Abstract

### Background

Different nutrient profiles (NPs) have been developed in Latin America to assess the nutritional quality of packaged food products. Recently, the Mexican NP was developed as part of the new warning label regulation implemented in 2020, considering 5 warning octagons (calories, sugar, sodium, saturated fats, and trans fats) and 2 warning rectangles (caffeine and non-nutritive sweeteners). The objective of this cross-sectional study was to evaluate the Mexican NP and other NPs proposed or used in Latin America against the Pan American Health Organization (PAHO) model.

### Methods and findings

Nutrition content data of 38,872 packaged food products available in the Mexican market were collected in 2016 and 2017. The evaluation of the Mexican NP, including its 3 implementation phases of increasing stringency (2020, 2023, and 2025), was conducted by comparing the percentage of products classified as "healthy" (without warnings) or "less healthy" (with 1 or more warnings), as well as the number and type of warnings assigned to food products, against the PAHO NP. Using the calibration method, we compared the classifications produced by the PAHO model against those produced by the NP models of Ecuador, Chile (3 phases), Peru (2 phases), Uruguay, and Brazil. Kappa coefficients and Pearson correlations were estimated, and proportion tests were performed. We found that the 3 implementation phases of the Mexican NP had near to perfect agreement in the classification of healthy foods (Mexico NP models: 19.1% to 23.8%; PAHO model: 19.7%) and a strong correlation (>91.9%) with the PAHO model. Other NPs with high agreement with the PAHO model were the Ecuador (89.8%), Uruguay (82.5%), Chile Phase 3 (82.3%), and Peru Phase 2 (84.2%) NPs. In contrast, the Peru Phase 1, Brazil, and Chile Phase 1 NP models had the highest

**Funding:** This work was funded by Bloomberg Philanthropies [grant number #43003]. (S.B). The funders had no role in study design, data collection and analysis, decision to publish, or preparation of the manuscript.

**Competing interests:** The authors have declared that no competing interests exist.

**Abbreviations:** FoPL, front-of-package labeling; NCD, noncommunicable disease; NP, nutrient profile; PAHO, Pan American Health Organization; WHO, World Health Organization.

percentage of foods classified as healthy (49.2%, 47.1%, and 46.5%, respectively) and the lowest agreement with the PAHO model (69.9%, 69.3%, and 73%, respectively). Study limitations include that warnings considered by the Mexican NP models were evaluated as if all the warnings were octagon seals, while 2 out of the 7 were rectangular warnings (caffeine and non-nutritive sweeteners), and that our data are limited by the quality of the information reported in the list of ingredients and the nutrition facts table of the products.

## Conclusions

The 3 implementation phases of the Mexican NP were useful to identify healthy food products. In contrast, the Peru Phase 1, Brazil, and Chile Phase 1 NP models may have limited usefulness for the classification of foods according to the content of ingredients of concern. The results of this study may inform countries seeking to adapt and evaluate existing NP models for use in population-specific applications.

## Author summary

### Why was this study done?

- Nutrient profile models that reflect the nutritional quality of food products are needed to help governments and decision-makers seeking to adapt simplified front-of-package labeling systems in their countries.

- The calibration of a nutrient profile aims to determine whether the nutrient profile model classifies foods correctly against reference methods, increasing the evidence supporting the model and improving its confidence. The Pan American Health Organization (PAHO) has developed a model that may work as a reference for Latin American countries.

- Several nutrient profiles have been developed and implemented in Latin America, with scarce evidence about their performance using representative databases of products of in the market.

- Even though the PAHO model was created for Latin American countries, it is relevant to adapt a nutrient profile model to local nutrition policies and epidemiology contexts. Recently, Mexico developed its own nutrient profile model based on the PAHO and Chile Phase 3 models, nutritional scientific population-based recommendations, and a feasibility analysis of products sold in the Mexican market.

### What did the researchers do and find?

- We compared the classification of a sample of more than 36,000 unique food products available in the Mexican market according to nutrient profile models from 6 countries (Mexico, Ecuador, Chile, Peru, Uruguay, and Brazil) against the PAHO model.

- According to the PAHO model, the Mexican Phase 3 nutrient profile performed best in identifying unhealthy products based on their content of energy, sugar, saturated fat, trans fat, sodium, non-nutritive sweeteners, and caffeine. This model had a strong correlation with the PAHO model (>91.9%).

## What do these findings mean?

- Our findings add support to the relevance of using the Mexican nutrient profile to evaluate the nutritional quality of food products as a basis for the prevention of obesity and chronic diseases related to diet.

- Our study has some limitations related to the quality of the information reported on the packages of the products included.

## Introduction

A nutrient profile (NP) is a tool that classifies foods and beverages according to their nutritional composition, e.g., whether products contain excessive amounts of ingredients of concern (sugar, sodium, and saturated and trans fats) and calories [1,2]. This tool allows the formulation and application of strategies related to the prevention and control of obesity and overweight [3,4], such as the use of front-of-package labels on processed foods, regulations for health or nutrition claims, regulation of unhealthy food marketing to children, food taxes, and restrictions on the foods and beverages available or sold in and outside schools [5,6].

Front-of-package labeling (FoPL) is a cost-effective strategy to promote healthy purchase decision-making in the population [3]. Recently, as a response to the growing epidemic of overweight and obesity in the Latin American region [7], several countries have adopted warning label systems. These are generally implemented in progressive phases to give the food and beverage industry the opportunity to reformulate their products in order to design new ones that do not exceed established thresholds for ingredients of concern [8]. For example, in 2016, Chile implemented warning labels for the first time in a 3-phase scheme (2016, 2018, and 2019) [9–11]. Chilean warning labels consist of black octagons with the legend "High in..." displayed on the front of the package for products with unhealthy levels of sugar, sodium, saturated fats, and/or calories [10]. Similar warning label systems have been adopted or are being considered in Peru (sugar, saturated fat, trans fat, and sodium), Uruguay (sugar, total fat, saturated fat, and sodium), Argentina (sugar, saturated fat, total fat, sodium, calories, non-nutritive sweeteners, and caffeine), and Brazil (sugar, saturated fat, and sodium) [12–15]. In 2014, Ecuador used its own NP to implement a traffic light system indicating whether a product contains relatively low (green), average (yellow), or high (red) levels of ingredients of concern (sugar, saturated fat, and sodium) [16]. In 2020, Mexico adopted new mandatory FoPL with warning octagons with the legend "Excess..." for calories and ingredients of concern (e.g. sugar, sodium, saturated fats, and trans fats) and warning rectangles for products that include added caffeine and non-nutritive sweeteners, along with the statement "avoid/not recommended in children" [17]. These national FoPL systems are implemented along with their own NP models, generally involving a 2- or 3-phase progressive implementation scheme, and some of them are based on the Pan American Health Organization (PAHO) NP model [17].

The PAHO model provides regional criteria for acceptable amounts of ingredients of concern (sodium, sugar, total fat, saturated fat, trans fat and non-nutritive sweeteners) in Latin America. However, NP models underpinning FoPL systems should be relevant for the national or regional food supply [18]. Therefore, regional criteria proposed by the PAHO model should be carefully adapted to ensure that the FoPL system is sensitive enough to classify products according to their healthfulness [18]. Classifications produced by the adapted NP models should also align with national dietary guidelines. In Mexico, sugar-sweetened beverages,

particularly cola drinks, are partly responsible for the high prevalence of obesity and diabetes [19]; these drinks are consumed at high levels by the population, including children [20]. Therefore, adaptations of the PAHO model for use in Mexico included additional thresholds for calories and a warning for added caffeine, intake of which is not recommended among children [21]. Other adaptations included removing the total fat threshold of the PAHO model, since total fat includes healthy fatty acids, which are not common in the Mexican diet and whose intake needs to be promoted. The sodium threshold proposed by PAHO was also adapted considering the national market share in Mexico. In accordance with the Mexican FoPL regulation, which involves 3 implementation phases (2020, 2023, and 2025), 3 NPs were proposed, with stricter nutrient criteria for each progressive phase [8,17].

According to the World Health Organization (WHO), the validation of a NP "considers different methods aimed at answering the question of whether the nutrient profile model classifies foods correctly" [2]. Validation is also needed to increase the evidence supporting the model, and hence improve confidence in the model [1]. There are different approaches to evaluating a NP model, involving calibration, construct validity, assessment of predictive validity against health outcomes in individuals, and experimental studies [22–24]. The calibration approach involves comparing the classifications produced by a NP model against those from another designed for similar purposes [2]. Currently, there is no gold standard for classifying the NP of a food product. However, the PAHO model has been adopted as a reference for comparing new NPs in various Latin American studies [10–13]. Thus, for Latin American countries, the PAHO model may work as a reference NP model [1]. The PAHO model was developed without food and beverage industry interference, and was the result of rigorous work by an Expert Consultation Group based on scientific evidence [4]. It is also based on the WHO population nutrient intake goals to prevent obesity and related chronic noncommunicable diseases (NCDs) [25], and considers the updated goals of the WHO expert consultations on maximum recommended intake of ingredients of concern: sugar, fats, and sodium [26].

The objective of this study was to evaluate the Mexican NP by comparing the percentage of products classified as "healthy" and "less healthy" (e.g., when product had 1 or more warnings), as well as the number and type of warnings assigned to food products, against the PAHO model using the calibration method. We also compared the classifications produced by the PAHO model against those from the NP models of Chile, Ecuador, Peru, Uruguay, and Brazil.

## The Mexican NP model

The Mexican NP model (Table 1) was developed based on the PAHO model and the Chilean NP, as well as on a feasibility analysis in more than 36,000 products retailed in Mexico. A brief description of the foundations of the Mexican NP is provided below.

The PAHO model proposes thresholds for 6 ingredients of concern (e.g., sugars, total fat, saturated fat, trans fat, sodium, and non-nutritive sweeteners), based on the population nutrient intake goals for preventing diet-related chronic diseases. These goals do not include a threshold for calories but are based on the energy contribution of each ingredient of concern. The PAHO model considers specific thresholds for each ingredient of concern based on its proportional contribution to the energy of the product, but it does not have a threshold for overall calories [17,25,26]. Hence, the Chilean NP model threshold for "high in calories" was used for beverages and solids in the Mexican NP model. The Chilean threshold for calories was determined according to the energy content in natural foods and beverages. For solids, the calorie threshold corresponds to the 90th–95th percentile value of the energy distribution for 100 g of natural foods, based on a food composition database [27]; this value has also been

**Table 1. Summary of nutrient profile (NP) models examined.**

| Country or entity | Implementation phases | Energy | Total sugar | Free or added sugars | Total fat | Saturated fat | Trans fat | Sodium | Other ingredients |
|---|---|---|---|---|---|---|---|---|---|
| PAHO | 1 (recommendation) | | | ✓ | ✓ | ✓ | ✓ | ✓ | Non-nutritive sweeteners |
| Mexico | 3 (2020, 2023, and 2025) | ✓ | | ✓ | | ✓ | ✓ | ✓ | Non-nutritive sweeteners, added caffeine |
| Chile | 3 (2016, 2018, and 2019) | ✓ | ✓ | | | ✓ | | ✓ | |
| Ecuador | 1 (2014) | | ✓ | | ✓ | | | ✓ | |
| Peru | 2 (2019 and 2022) | | ✓ | | | ✓ | ✓ | ✓ | |
| Uruguay | 1 (2020) | | ✓ | | ✓ | ✓ | | ✓ | |
| Brazil | 1 (proposal) | | | ✓ | | ✓ | | ✓ | |

PAHO, Pan American Health Organization.

used to determine energy-dense foods [28]. For beverages, the Chilean model considers the energy content of plain milk per 100 mL (70 kcal) as the reference [11,29]. Nevertheless, according to this threshold some dairy beverages are classified as "high in calories" and "high in sugars," while soft drinks are only labeled as "high in sugars" creating missunderstanding that soft drinks are healthier than dairy beverages [30]. This is controversial since all calories in soft drinks come from added sugar, while for sweetened dairy beverages, other nutrients also account for total calories (e.g., lactose, proteins, and fat). Indeed, soft drinks have been directly linked with the development of some NCDs, such as obesity and diabetes [26,31]. To solve this limitation, the Mexican NP model proposes a new cutoff point for the "Excess calories" threshold in beverages. This criterion was developed considering 2 population scientific bases. The first scientific base was the WHO recommendation to reduce daily intake of free sugars to less than 10% of total energy intake (equivalent to less than 50 g of sugars per day on a 2,000-calorie diet.) [32,33]. The second scientific base was the daily average beverage intake of 2,520 mL/day; this volume was established by the Beverage Consumption Recommendations for the Mexican Population, considering a diet that provides 2,200 calories and an adequate intake of all essential nutrients [34]. By dividing the WHO recommended daily intake of free sugars by the daily average beverage intake recommendation in Mexico, a standardized rate of 1.98 g of sugar per 100 mL was obtained. For ease of calculations, this figure was rounded up into 2 g/100 mL, equivalent to 8 kcal per 100 mL (or approximately 5 g of sugar [1 teaspoon] per 250 mL), and set as the final threshold for "Excess calories" for the Mexican NP model. However, for the first implementation phase of the Mexican FoPL system, a limit of <10 kcal per 100 mL was established (e.g., products with ≥10 kcal/100 mL receive the label "Excess calories"). This threshold was established by the Mexican authorities in order to provide manufacturers enough time to reformulate their products.

Thresholds for "Excess sodium" in solid and liquid products were also established. Based on available PAHO recommendations at the time [4], a threshold of >1 mg of sodium per 1 kcal was initially established for solid and liquid products [35]. However, the proposed threshold had the limitation that it classified some low-calorie beverages with minimum amounts of added sodium (e.g., diet soda) as "Excess sodium". For example, a low-calorie beverage with 5 kcal and 10 mg of sodium would exceed this threshold. Therefore, we developed a new threshold for non-caloric beverages based on the maximum content of sodium in beverages available in the Mexican market. First, all beverages not exceeding the sodium threshold (>1 mg/kcal) proposed by PAHO were selected (n = 425, 85% were carbonated beverages). Then, we analyzed the sodium content per 100 mL of these beverages, finding that the maximum content

was of 45 mg [31]. We established this threshold (45 mg of sodium per 100 mL) as the criterion for the "Excess sodium" label for non-caloric beverages. This threshold is similar to the 2020 PAHO threshold for sodium for ultra-processed and processed drinks that provide no energy (40 mg/100 mL) that was published later on [35]. Based on the updated PAHO thresholds, we decided to additionally consider 300 mg of sodium per 100 g or 100 mL as the sodium threshold for the rest of food products [35]. This threshold would allow a more stringent identification of products with excess of sodium, especially desserts, snacks, and sauces/condiments.

## Methods

### Food products retailed in the Mexican market

Public databases containing information on nutrient composition and ingredient information for branded foods and beverages in Mexico were not available at the time of the study. Therefore, nutrient composition data on 38,872 packaged food products retailed in the Mexican market were collected from 23 January 2016 to 15 December 2017 following standardized procedures for measuring packaged foods and beverages according to Kanter et al., consisting of photographing selected product packaging at points of sale, followed by information download and data entry at the office using the software Research Electronic Data Capture (REDCap, Vanderbilt University, Nashville, TN, US) (see S1 Text) [36].

The protocol of this study was approved by the research, ethics, and biosafety committees of the Mexican National Institute of Public Health (approval number: 1530). This study was part of the study INFORMAS, monitoring and benchmarking food environments globally [37].

Fieldworkers were trained nutritionists who attended a 1-month workshop [38] that consisted of training and experiential learning on food composition, food labeling, and food promotion of packaged foods. In addition, fieldworkers received a 1-day photography workshop led by a professional photographer, and half-week fieldwork training in a real supermarket. To standardize data collection processes, fieldworkers were provided with 50 products from different categories and of varying shapes and sizes. Products were placed on a table in a large classroom and fieldworkers were instructed to photograph the products, download images to their computers, and capture data using REDCap [39]. Data captures were compared, and fieldworkers were considered trained when inter-rater reliability was >80%.

Data were collected in 8 cities purposively selected to capture the diversity of foods available in urban areas in the country and to compare different regions of Mexico (e.g., the northern part of the country has a wide variety of imported food products from the United States of America). These cities included the 4 cities with the highest population density and economic relevance in the country (Mexico City, Guadalajara, Monterrey, and Queretaro) as well as other smaller cities (Cuernavaca, Baja California, Saltillo, and Ciudad Juárez). The study is reported according to STROBE (https://www.strobe-statement.org) (see S2 Text) [40].

We selected a convenience sample of 127 and 136 retail stores in 2016 and 2017, respectively. Retail stores were geographically located in areas with high population densities. First, the National Statistical Directory of Economic Units [41] was used to identify all economic units classified as "retail trade in supermarkets" within each city. Of these, all hypermarkets, supermarkets, and convenience stores were initially selected. Then, a 1,000-m buffer around each establishment was drawn, and the total population within the buffer according to the 2010 census was calculated [42]. Subsequently, the distance to the nearest retail store was calculated. All retail stores with a population density of >20,000 inhabitants within the 1,000-m buffer and with a distance of >1,500 m to the nearest retail store were included. Regardless of the above criteria, all membership food stores from the top grocery retailers in Mexico were

included because these establishments market select brand products that are available only in these stores. The included food retailers altogether represent more than 70% of the market share in the country [43]. The final number of stores visited in each city varied from 3 in Cuernavaca to 65 in Mexico City.

Within each store selected, fieldworkers photographed all packaged food products available at the time, including all food categories and all brands. This strategy was followed because we were interested in evaluating differences in the availability of products according to the socioeconomic status of the neighborhoods in which stores were located. Subsequently, duplicate products were removed. The data collected by the fieldworkers included product information (e.g., company, brand), net content, price, nutrient facts panel information, ingredients list, health and nutrition claims, and FoPL, from photos of all sides of the packaging. Nutrition information was recorded and in the case of reconstituted products, the "as consumed" information was retrieved from the photographs of the products. Field supervisors revised the completeness and accuracy of the captured data.

The database was then transferred to Stata (version 14, StataCorp, College Station, TX, US) format to be reviewed and cleaned. Foods and beverages were classified into 23 groups that commonly include processed or ultra-processed products and have been previously used for the discussion of food nutrition policies: eggs, legumes, soups, potatoes/yams, marine products, packaged salads, cereal/grains, combination dishes, sugars/sweets, nuts/seeds, vegetables, meat/poultry, desserts, sauces/condiments, bakery products, dessert toppings/fillings, snacks, miscellaneous items, fats/oils, fruit/fruit juices, dairy beverages, and non-dairy beverages (S1 Table) [44].

## Latin American NPs

This study did not have a prespecified analysis plan. Food and beverages were classified according to different NPs from 6 countries (Mexico, Chile, Uruguay, Brazil, Peru, and Ecuador) considering different stages of implementation, from the first, or most permissive, to the last, or definitive one, as well as the PAHO NP model (Table 1). Thus, a total of 12 NPs were included in the study. Of these, 9 corresponded to approved NP models for octagon warning labeling systems (Mexico [$n$ = 3] [17], Chile [$n$ = 3] [9,45,46], Peru [$n$ = 2] [12,47], and Uruguay [$n$ = 1] [13]), 1 to the multiple-traffic-light system in Ecuador ($n$ = 1) [46,48,49], and 1 to the proposed NP in Brazil as part of a labeling system similar that implemented in Canada ($n$ = 1) [50]. Specifications for each NP were retrieved from official records of the Ministry of Health of each country. Table 1 summarizes the NP models examined and the ingredients of concern evaluated. The characteristics of the Latin American NP models and their detailed cutoff points are shown in S2 and S3 Tables, respectively.

As suggested by the PAHO model, food and beverages were categorized according to the NOVA food classification [4,44]. Of the 38,872 products included in the dataset, a total of 2,028 were excluded from the analysis because information was not available for energy or some of the ingredients of concern evaluated or because they were baby food. For all the assessed NP models, unprocessed or raw products were classified as compliant with the NP criteria.

The NPs of processed foods were independently calculated by 2 researchers using algorithms generated in Stata. The results obtained by each researcher were compared. Any disagreements were discussed and resolved until consensus was reached. Then, the NPs of a random sample of 30 products were manually calculated and compared against those calculated by Stata algorithms. This process was repeated for each NP until results (e.g., Stata algorithms versus manual calculations) matched 100%.

## Calibration of NP models

The 3 Mexican NPs (implementation phases 1–3), as well as the NP models from Uruguay, Ecuador, Brazil, Peru (phase 1 and 2), and Chile (phases 1–3), were assessed using the calibration method against the PAHO model. For this purpose, we used the following indicators.

**Healthy versus less-healthy products.** Food products were categorized as "healthy" (e.g., when the product was classified as having no warnings) or "less healthy" (e.g., when the product displayed 1 or more warning labels). Kappa coefficients were estimated to evaluate the agreement and consistency in the classifications across NPs. Kappa coefficients were used to categorize agreement as follows: 0.01–0.20, slight; 0.21–0.40, fair; 0.41–0.60, moderate; 0.61–0.80, substantial; 0.81–0.99, near perfect.

**Types of warnings.** Food products were classified according to the ingredient-specific threshold of each NP. For this indicator, the PAHO model was considered the reference for sugar, sodium, and saturated fat, while Chile Phase 3 was considered the reference for calories. Given that not all NPs evaluated the same nutrients, we evaluated calories for the Mexico and Chile NP models; sugar (total or free) and sodium for all NP models; total fat for the PAHO, Ecuador, and Uruguay NP models; and saturated fat for all NP models except Ecuador. Kappa coefficients were used to assess the agreement of each NP with the reference NP in the percentage of products classified with each ingredient-specific warning.

**Number of warnings.** Products were classified according to the number of warnings assigned by each NP. If the product complied with all the criteria (categorized as healthy) or if it was not ranked by the NP (e.g., because it was a raw or an unprocessed product), it was classified as having 0 warnings. Further, the number of warnings assigned by each NP model was estimated as follows. For the Mexico Phase 3 NP model, products were classified as having 0 to 7 warning labels (5 octagons for calories and 4 ingredients of concern plus 2 rectangles for child health protection—caffeine and non-nutritive sweeteners). For the PAHO model, products were classified as having 0 to 6 warnings, according to the 6 ingredients of concern of the model. For the Chile, Peru, and Uruguay NP models, products were classified as having 0 ("no warnings") to 4 warnings overall. For the Ecuador NP model, ingredients of concern labeled yellow or red were considered warnings. For example, if a product had 2 of the 3 possible ingredients (total fat, sugar, and sodium) in green and 1 in yellow or red, the product was classified as having 1 warning. Thus, a food product could have 0, 1, 2, or 3 warning labels.

To evaluate the agreement and correlation between the PAHO model and the Mexico Phase 3 NP, we recoded the number of warning labels assigned by these 2 models as ranging from 0 to 5 or more warning labels (out of 6 or 7 possible warnings, respectively). Pearson correlation coefficients, kappa coefficients, and correlation tests were used to compare the number of warnings between these 2 models. The same approach was used to compare the number of warning labels by food group. Additionally, we compared the number of warnings among all NP models using a similar approach.

## Results

### Agreement in the percentage of products classified as "healthy" and "less healthy"

Fig 1 shows the percentage of foods classified as healthy (e.g., with no warnings) and less healthy (e.g., with 1 or more warnings) by each NP model, as well as the agreement between the PAHO model and the rest of the NP models. Overall, 19.9% of products were classified as healthy according to the PAHO model. The Mexican Phase 1 (19.4%), Phase 2 (20.4%), and Phase 3 (24%) NPs classified a similar percentage of packaged foods as having no warnings,

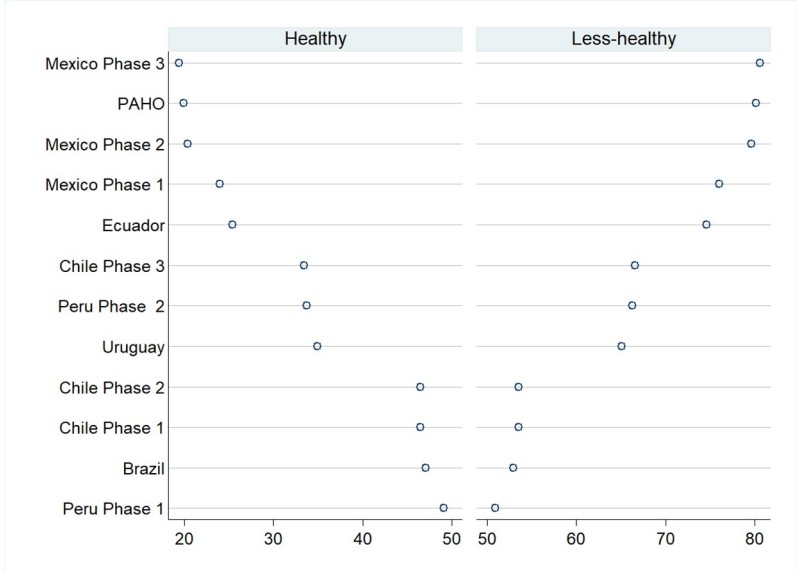

**Fig 1. Percentage of products classified as "healthy" and "less healthy" by the PAHO model and Latin American NPs ($n$ = 36,844 unique packaged products).** Products with no warning labels were classified as healthy; products with 1 or more "high in. . ." warnings were classified as less healthy. For the Ecuador NP model, products with 1 or more warnings of moderate (yellow) or high (red) content of nutrients of concern were classified as less healthy. Percent agreement (A) was assessed using kappa coefficients ($k$): 0.01–0.20, slight; 0.21–0.40, fair; 0.41–0.60, moderate; 0.61–0.80, substantial; 0.81–0.99, near perfect. PAHO (reference); Mexico Phase 3 ($k$ = 0.861; A: 95.6%); Mexico Phase 2 ($k$ = 0.764; A: 95.3%); Mexico Phase 1 ($k$ = 0.764; A: 91.9%); Ecuador ($k$ = 0.764; A: 89.8%); Uruguay ($k$ = 0.572; A: 82.5%); Chile Phase 3 ($k$ = 0.557; A: 82.3%); Peru Phase 2 ($k$ = 0.604; A: 84.2%); Chile Phase 2 ($k$ = 0.479; A: 77.7%); Chile Phase 1 ($k$ = 0.372; A: 69.9%); Brazil ($k$ = 0.431; A: 73%), Peru Phase 1 ($k$ = 0.379; A: 69.3%). All comparisons were statistically significant ($p < 0.05$). NP, nutrient profile; PAHO, Pan American Health Organization.

having a high agreement with the PAHO model ($k > 91.9\%$, substantial to near perfect). Similarly, the Ecuador (89.8%, $k$ = 0.707), Uruguay (82.5%, $k$ = 0.572), Chile Phase 3 (82.3%, $k$ = 0.557), and Peru Phase 2 (84.2%, $k$ = 0.604) NPs showed moderate to high agreement with the PAHO model. In contrast, the Chile Phase 1, Brazil, and Peru Phase 1 NP models had the highest percentage of foods classified as healthy (46.5%, 47.1%, and 49.2%, respectively) and the lowest agreement with the PAHO model ($<70\%$, $k < 0.432$, moderate) (Fig 1). All comparisons were statistically significant ($p < 0.05$).

## Agreement in the types of warnings

Table 2 shows the agreement in the types of warning labels assigned to food products between the PAHO model and the other NP models. The highest percentage of products with "high in sugar" warning was found for the PAHO model and Mexico Phase 2 and Phase 3 NP models (40.4% for the 3 models). The agreement between the PAHO model and these Mexican NP models was near perfect. The Peru Phase 1 and Chile Phase 1 NP models had the lowest percentage of products with warnings for sugar (approximately 28%); the Peru Phase 1 NP had the lowest agreement with the PAHO model for this ingredient ($k$ = 0.643, 83.6%).

The PAHO model classified 40.1% of products as high in sodium. The Ecuador (47.3%) and Mexico Phase 3 (43.4%) NP models had a slightly higher percentage of products labeled with this warning. The agreement between the NP models the PAHO model was moderate (78.7%, $k$ = 0.519) for the Ecuador NP model and near perfect for the Mexico Phase 3 NP

**Table 2. Agreement of Latin American NP models with the PAHO model for the percentage of products with warnings for ingredients of concern, and with the Chilean model for the percentage of products with the warning "high in calories" (n = 36,844 unique packaged products).**

| NP model | High in sugar | | | High in sodium | | | High in saturated fat | | | High in calories | | | High in total fat | | | High in trans fat* | | | With non-nutritive sweetener | | | With added caffeine | | |
|---|---|---|---|---|---|---|---|---|---|---|---|---|---|---|---|---|---|---|---|---|---|---|---|---|
| | Freq | k | %A | Freq | k | %A | Freq | k | %A | Freq | k | %A | Freq | k | %A | Freq | k | %A | Freq | k | %A | Freq | k | %A |
| PAHO | 40.4 | Ref. | Ref. | 40.1 | Ref. | Ref. | 34.7 | Ref. | Ref. | NA | NA | NA | 40.1 | Ref. | Ref. | 1.0 | Ref. | Ref. | 12.7 | Ref. | Ref. | NA | NA | NA |
| Chile Phase 3 | 36.8 | 0.885 | 94.5 | 31.1 | 0.661 | 82.2 | 20.2 | 0.615 | 84.3 | 38.9 | Ref. | Ref. | | | | | | | | | | | | |
| Chile Phase 2 | 32.7 | 0.815 | 91.3 | 26.0 | 0.606 | 82.2 | 20.2 | 0.615 | 84.3 | 36.7 | 0.954 | 97.8 | | | | | | | | | | | | |
| Chile Phase 1 | 27.6 | 0.701 | 86.3 | 15.1 | 0.365 | 73.2 | 18.6 | 0.588 | 83.4 | 32.4 | 0.859 | 93.5 | | | | | | | | | | | | |
| Mexico Phase 3 | 40.4 | 1.000 | 100.0 | 43.4 | 0.819 | 91.2 | 33.9 | 0.977 | 98.9 | 48.3 | 0.750 | 87.6 | | | | 1.0 | 0.999 | 99.9 | 12.7 | 0.999 | 99.9 | 0.8 | | |
| Mexico Phase 2 | 40.4 | 1.000 | 100.0 | 41.7 | 0.790 | 89.9 | 25.8 | 0.791 | 91.1 | 44.5 | 0.824 | 91.4 | | | | 1.0 | 0.999 | 99.9 | 12.7 | 0.999 | 99.9 | 0.8 | | |
| Mexico Phase 1 | 40.0 | 1.000 | 100.0 | 32.8 | 0.669 | 84.6 | 25.8 | 0.791 | 91.1 | 44.2 | 0.830 | 91.7 | | | | 1.0 | 0.999 | 99.9 | 12.7 | 0.999 | 99.9 | 0.8 | | |
| Peru Phase 2 | 38.6 | 0.807 | 90.8 | 31.1 | 0.701 | 86.2 | 27.1 | 0.748 | 89.2 | | | | | | | 1.9 | 0.706 | 96.9 | | | | | | |
| Peru Phase 1 | 27.9 | 0.643 | 83.6 | 13.9 | 0.375 | 73.2 | 22.6 | 0.696 | 87.4 | | | | | | | 1.9 | 0.706 | 96.9 | | | | | | |
| Ecuador | 32.7 | 0.767 | 89.1 | 47.3 | 0.519 | 78.7 | | | | | | | 23.1 | 0.606 | 82.3 | | | | | | | | | |
| Uruguay | 35.0 | 0.864 | 93.6 | 27.8 | 0.659 | 84.5 | 22.4 | 0.633 | 84.8 | | | | 29.1 | 0.639 | 83.4 | | | | | | | | | |
| Brazil | 27.8 | 0.721 | 87.2 | 21.3 | 0.563 | 55.7 | 23.7 | 0.722 | 88.3 | | | | | | | | | | | | | | | |

%A, percent agreement; Freq, frequency; k, kappa coefficient; NA, not applicable; NP, nutrient profile; PAHO, Pan American Health Organization. Kappa coefficients were used to categorize agreement as follows: 0.01–0.20, slight; 0.21–0.40, fair; 0.41–0.60, moderate; 0.61–0.80, substantial; 0.81–0.99, near perfect. All comparisons were statistically significant ($p < 0.05$). For "high in calories," Chilean Phase 3 NP was the reference.

*For Peru the warning label threshold is any amount of trans fat added to the product.

model (91.2%, $k = 0.819$). The Peru Phase 1 and Chile Phase 1 NP models had the lowest percentage of warnings for sodium (13.9% and 15.1%, respectively).

The PAHO (34.7%) and Mexico Phase 3 NP (33.9%) models had the highest percentage of products with "high in saturated fat" warning; their agreement was near perfect (98.9%, $k = 0.977$). The Chile Phase 1 NP model had the lowest agreement with the PAHO model for this ingredient (83.4%, $k = 0.588$, moderate), with 18.6% of products classified as excessive in saturated fat.

For calories, the Chile Phase 3 NP was the reference model, and classified 38.9% of products as high in calories. The Mexico Phase 1 NP model had the highest agreement (92.6%) with the Chile Phase 3 NP, classifying 44.3% of the products with this warning. Meanwhile the Mexico Phase 3 NP classified 48.4% of the products as excessive in calories, in near perfect agreement with the Chile Phase 3 NP (87.6%, $k = 0.750$). No other NP models considered this criterion.

For total fat, the Ecuador and Uruguay NP models had substantial agreement with the PAHO model ($k = 0.606$ and $0.639$, respectively). Finally, near perfect agreement ($k = 0.999$) between the Mexico NP models and the PAHO model was observed for trans fat (1.0%) and non-nutritive sweeteners (12.7%) (Table 2). The Peru NP had substantial agreement ($k = 0.706$) with the PAHO model for trans fat. Added caffeine was only evaluated for the Mexico NP models (Table 2). All comparisons were statistically significant ($p < 0.05$).

## Agreement in the number of warnings

Fig 2 and S4 Table show the percentage of products with 0 to 5 or more warning labels, and the agreement and correlation between the PAHO model and the Mexico Phase 3 NP in the number of warning labels assigned to products, overall and by food group. Overall, 57.4% of the products were classified as having the same number of warnings by the 2 NP models, and the correlation between them was of 0.813. Across food groups, the agreement between these 2 NP models was highest (e.g., $k$ was near perfect) for fruit/juices (95.8%), followed by

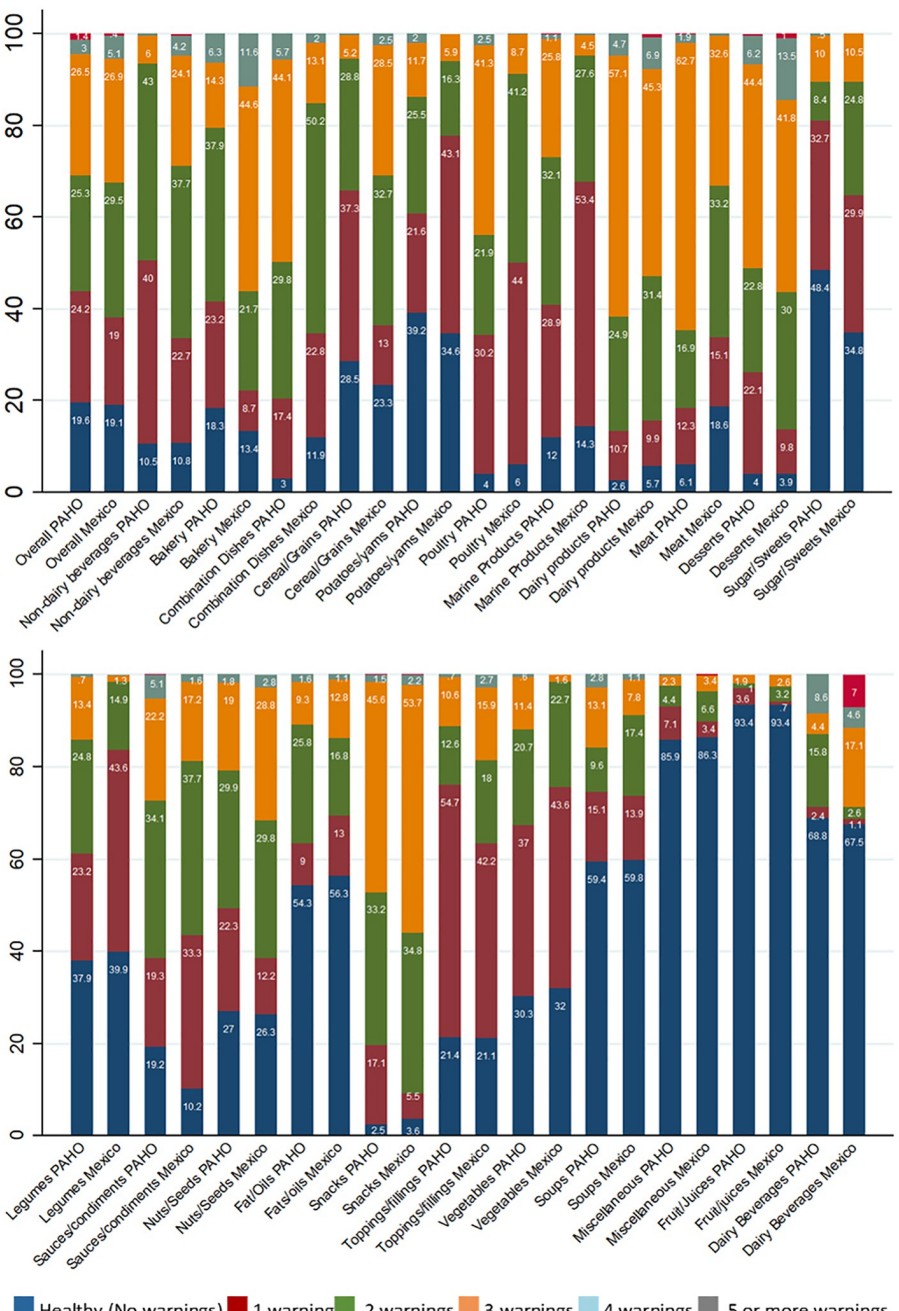

**Fig 2. Percentage of products with 0 to 5 or more warning labels assigned by the PAHO model and the Mexico Phase 3 nutrient profile, overall and by food group (*n* = 36,844 unique packaged products).** All comparisons were statistically significant (*p* < 0.05). PAHO, Pan American Health Organization.

miscellaneous items (91.1%) and soups (81.9%). For toppings/fillings, fats/oils, vegetables, and snacks, agreement between the PAHO model and the Mexico Phase 3 NP was between 74.5% and 77.3%. The food groups with the lowest agreement (e.g., below 70%) between these 2 models were non-dairy beverages (31.9%), bakery products (32.1%), and combination dishes (37.5%): The Mexican NP identified a higher percentage of beverages and bakery products as

having high content of ingredients of concern, while the PAHO NP model identified a higher percentage of combination dishes as being high in ingredients of concern (Fig 2; S4 Table).

S1 Fig summarizes the overall distribution of products according to the number of warning labels assigned to food products by all the included NP models. The Mexico Phase 3 NP model had the highest percentage of food products classified with 3 (26.8%) and 4 or more warnings (5.5%), while the Peru Phase 1 NP model had the lowest percentage of products with 2 (14.1%) and 3 warnings (0.9%).

S5 Table summarizes the Pearson correlation coefficients and kappa coefficients across all the included NP models. Correlations between all NP models varied widely. The highest correlation for the number of warnings assigned to food products was found between the Brazil and Ecuador NP models ($r = 0.930$), and the lowest correlation was found between the PAHO model and the Chile Phase 1 NP model ($r = 0.486$). Similarly, the NP model with the highest agreement with the PAHO model regarding the classification of products as healthy and less healthy was the Mexico Phase 3 NP model (95.6%, $k = 0.861$), whereas the lowest agreement was observed for the Peru Phase 1 (69.3%, $k = 0.379$) and Chile Phase 1 NP models (69.6%, $k = 0.372$). All comparisons were statistically significant ($p < 0.05$).

## Discussion

In this study the Mexican NP model was evaluated through the calibration method against the PAHO model using a large sample of Mexican packaged foods and beverages. Results indicate high agreement and correlation in study outcomes between the 3 implementation phases of the Mexican NP model and the PAHO model. Further, results also provide information on the comparability of other NP models proposed or used in Latin America as the underlying criteria for FoPL schemes, showing a wide variability in their ability to identify products with high amounts of ingredients of concern, despite most of them being based on the PAHO model. These results underscore the relevance of choosing and adapting a NP model to local nutrition policies.

### Agreement in the classification of healthy versus less healthy products

To date, scarce evidence is available regarding calibration of Latin American NP models. In a study conducted in a sample of products from Brazil, Duran et al. [51] classified 38% of the products as healthy according to the PAHO model, while more than half of the products were classified as healthy by the Chile Phase 3 (58%) and Brazil (55%) NP models. In our study, the PAHO model and the 3 phases of the Mexican NP model classified the lowest percentage of products as healthy (around 20%), whereas almost half of the products were classified as healthy by the Brazil model, and around a third were classified as healthy by the Chile Phase 3 model. It is important to note that the profile used by Duran et al. for Brazil was a more stringent version [52] than the recently approved NP used in our study. Hence, one would expect a higher percentage of products to be classified as healthy according to the less stringent version used in our study than the one used by Duran et al. [51]. However, we found a lower percentage of products classified as healthy in our database by using the less stringent version. This inconsistency may be explained by differences in the overall healthiness of products included in the 2 studies. Products in our database were less healthy than the ones included in Duran et al.'s study, as suggested by the lower percentage of healthy products in our study compared to Duran et al.'s study according to both the PAHO (38% versus 20%) and the Chile Phase 3 models (58% versus 33%) [51].

Overall, our results indicate that the 3 implementation phases of the Mexican NP are useful to classify food products according to their nutritional quality, when compared to the PAHO

model. However, the Mexican NP was stricter than other Latin American NPs, and, for some food and beverage groups, it was even stricter than the PAHO model. Differences may be explained by the additional threshold for calories in the Mexican NP. Due to the alarming epidemic of obesity and diabetes in Latin American countries, a warning label system with a stringent NP like the PAHO model or the Mexican NP is considered ideal to inform consumers in a clear and simple way, especially because packaged processed foods usually have high amounts of calories, added sugars, unhealthy fats, and sodium that the general population is unaware of. In this context, it has been proposed that a NP model classifying more than half of products as healthy may have a reduced ability to improve consumers' dietary behaviors [53] and to promote product reformulation [54] aimed at reducing the content of nutrients of concern. According to our study, the Peru Phase 1, Brazil, and Chile Phase 1 and Phase 2 NP models classified around half of the products as healthy. Nevertheless, most of these models were transitional, except for the Brazil NP, which was the final one. The fact that Brazil's NP model showed the lowest agreement with the PAHO model among the NPs studied may have important implications for the ability of consumers in the Brazilian population to correctly classify the nutritional quality of food products.

## Agreement in the type of warnings

Our results are in line with the results of the study by Duran et al. indicating high agreement between the PAHO model and the Chilean Phase 3 NP in the percentage of food products exceeding the thresholds for sugar, sodium, and saturated fat [51]. As for the sugar threshold, Duran et al. classified 51.8% of products as high in sugar with the PAHO model, 34.5% with Brazil's model, and 38.1% with the Chile Phase 3 model. Similar percentages were observed in our study for these models (PAHO, 40.9%; Brazil, 28.1%; Chile Phase 3, 37.3%) as well as for the Mexico Phase 3 NP. Similarly, and in line with Duran et al. [51], between 30% and 40% of food products exceeded the sodium threshold according to the PAHO model, Chile NP models, and Mexico Phase 3 model. Finally, according to Duran et al. [51], 35.4% of the products were classified as high in saturated fats with PAHO NP model, 29% with the Brazilian model, and 20.4% with the Chilean model. In concordance with our study, the PAHO model and the Mexican Phase 3 NP classified around 30% of products as having excess saturated fat, while the Chile and Brazil NP models classified around 20%.

The Mexican NP model considers a threshold for calories, which is not included in the PAHO model. Our results indicate that this threshold is highly correlated with that of the Chile NP model. This is relevant information for the Mexican context since sugar-sweetened beverages represent the "less healthy" food group with the highest daily consumption among Mexicans across age groups (ENSANUT 2020) [20]; furthermore, 40,842 deaths per year in Mexico are attributable to the consumption of these beverages [19].

Non-nutritive sweeteners are important ingredients to be warned about, since their consumption has been associated with the habitual use of sweet flavors (sugar-based or not), dysbiosis of the gut microbiome, weight gain, and higher risk for type 2 diabetes [55–58]. This criterion is evaluated by the PAHO model and the Mexico NP, based on the potential harmful effects of non-nutritive sweeteners among children [58], and our study shows high agreement between the models. Despite the fact that the original NP proposals in Chile, Peru [59], and Uruguay [60] also included a threshold for this ingredient, the final regulations dropped it due to food industry interference. In consequence, product reformulation in these countries consisted partly in replacing added sugar with non-nutritive sweeteners [61]. In comparison, product reformulation in Mexico has been achieved by reducing the amount of added sugar and including other novel ingredients, such as lactase for dairy beverages and allulose for

breakfast cereals. The health effects of these food industry responses are to be studied in future research.

The warning for added caffeine targeting children was included in the Mexican NP model based on the argument that caffeinated products may cause hyperactivity, insomnia, addiction, and increases in blood pressure among consumers [21,62]. According to our results, less than 1% of products have this warning, causing little impact on the classification agreement between this model and the PAHO and Chile NP models. Nonetheless, by including specific thresholds targeting children, the impact of the new warning labels may be maximized since studies in Chile suggest that young populations were the main drivers for change among Chilean families [63].

Finally, total fat is considered by only some NP models, because total fat includes healthy (e.g., monounsaturated and polyunsaturated fatty acids) and unhealthy (e.g., saturated or trans) fatty acids. Our results indicate high agreement in the percentage of products exceeding this threshold for the PAHO, Ecuador, and Uruguay NP models.

## Agreement in the number of warning labels

In our study, classifications varied widely when comparing the number of warnings across Latin American NP models. Results demonstrate that even small differences in the selection of calories and ingredients of concern included in a NP model may cause big differences in the classification of the products. Differences across NPs and the PAHO model are mostly due to the number of ingredients of concern evaluated by each. The Mexico NP had the highest number of thresholds established ($n = 7$), followed by the PAHO model ($n = 6$). In contrast, the Ecuador and Brazil NPs have the lowest number ($n = 3$). Nevertheless, overestimation of the percentage of healthy products was higher for Brazil than for Ecuador because Brazil's thresholds are more flexible. Differences in the number of warning labels may also be caused by the type of ingredient. For example, for the Mexico NP and the PAHO model, the sugar threshold considers the content of free sugars, while other NPs such as Chile and Peru, total sugars are considered. Free sugars are more useful than total sugars to identify sugars added to the products, since sugars other than monosaccharides or disaccharides are present in products in a natural manner, such as lactose and fructose. [58]. Similarly, the PAHO model and the Mexico NP models evaluate trans fat content in processed foods using a threshold based on the percentage of energy derived from trans fats (>1%), while Peru NP classifies products with a "Contains trans fats" warning when any of this ingredient is added to the product.

Results indicate that the Mexico Phase 3 NP had high agreement with the PAHO model for most food groups. However, disagreement (<40% agreement) was observed within specific food groups, including non-dairy beverages, combination dishes, and bakery products. Disagreement between these 2 NP models was observed in the percentage of products with 1 or more warnings, which may be explained by the "Excess calories" warning of the Mexican NP model and the "high in total fat" criteria or threshold of the PAHO model. For bakery products, the "Excess calories" warning of the Mexican NP model accounts for 1 additional warning in 77.4% of the products in this food group; similarly, the total fat criterion of the PAHO model accounts for 1 additional warning in 17% of these products. For combination dishes, a total of 23% of the products in this food group were labeled as having excess calories according to the Mexican NP model, whereas 75% were classified as high in total fat by the PAHO model. Finally, for non-dairy beverages, 55% of the products were labeled as having excess calories by the Mexican NP, while only 3% were labeled as high in total fat by the PAHO model. Despite disagreements between the models, it must be noted that for both models around 10% of the products were classified as having 0 warnings (e.g., healthy).

## Strengths and limitations

This study considered a large sample of packaged products retailed in Mexico collected in 2016 and 2017, before the implementation of the warning labels. Also, to our knowledge this is the first study comparing all NP models used in Latin America, filling the knowledge gap regarding the comparability of these NPs.

Nonetheless, results should be interpreted considering some limitations. First, warning rectangles (caffeine and non-nutritive sweeteners) considered by the Mexican NP model were evaluated as if all of the warnings were octagonal seals. Hence, the maximum number of warnings for the Mexican NP model was 7 (5 octagons and 2 rectangles). A similar approach was used when evaluating Ecuador's NP, by giving a similar interpretation to both the yellow and the red traffic lights. These decisions may have overestimated the strictness of the Mexico and Ecuador NP models. Nevertheless, rectangles and yellow warnings may have equal importance to octagons and red warnings, since they can be used to regulate advertising directed to children or to regulate the use of health and nutrition claims [64]. Second, our data were limited by the quality of the information reported in the list of ingredients and the nutrition facts table of the products. The regulation enforced during the data collection period allowed 0 values to be reported for energy and ingredients of concern when their content was very small (e.g., small products such as candies reported servings of 1 g of sugar per portion). This was more common in products with small portion sizes, opening the possibility for underreporting. Nevertheless, the Federal Commission for the Protection against Sanitary Risks (COFEPRIS by its acronym in Spanish) monitors and assesses the nutrition composition and labeling of packaged foods and beverages, preventing manufacturers from reporting incorrect information on the labels. Additionally, food manufacturers are required to submit a bromatology analysis of their product before it reaches the Mexican market. Due to that process, we reckon that the content declared is accurate in general. However, no national assessment of the accuracy of packaged food information has been published. As with all food composition databases, there is a risk of error. Under this assumption, our results are conservative, and the number of foods and beverages classified as less healthy could be higher.

## Conclusion

The 3 implementation phases of the Mexican NP were useful to identify healthy food products. In contrast, the Brazil NP model had the lowest correlation with the PAHO model, suggesting that this model may have limited usefulness for the classification of foods according to the content of ingredients of concern. Findings highlight the importance of examining warning label classifications in various ways, considering classification within food groups, as well as the types and numbers of warning labels, to observe differences between models. The results of this study may inform countries seeking to adapt and evaluate existing models for use in country-specific applications.

## Supporting information

**S1 Data.**
(DTA)

**S1 Fig. Classification of 36,844 unique packaged products according to 7 Latin American nutrient profiles.** Overall number of warnings includes warnings in captions and warnings in octagons for the Mexican (0 to 7 warnings), Uruguay (0 to 4 warnings), Chilean (0 to 4 warnings), and Peru (0 to 3 warnings) nutrient profile models, and yellow and red traffic lights for

Ecuador's nutrient profile model (0 to 3 warnings). PAHO, Pan American Health Organization. *All comparisons were statistically significant ($p < 0.05$).
(TIF)

**S1 Table. Grouping of packaged foods.** *The sample comprises unique products.
(DOCX)

**S2 Table. Latin American nutrient profiles.**
(DOCX)

**S3 Table. Detailed cutoff points of Latin American nutrient profiles.** All nutrient profiles were specified for 100 g or 100 mL.
(DOCX)

**S4 Table. Percentage agreement and correlation between the PAHO model and the Mexico Phase 3 nutrient profile in the number of warning labels assigned to products, overall and by food group ($n$ = 36,844 unique packaged products).**
(DOCX)

**S5 Table. Pearson correlations of the number of warnings and kappa coefficients of healthy or less healthy across Latin American nutrient profiles classifying 36,844 unique packaged products to assess construct/convergent validity.** Pearson correlation coefficient (P) for number of warnings (Mexico, 0 to 7; Chile, Uruguay, and Peru, 0 to 4; Ecuador, 0 to 3; Brazil, 0 to 3; PAHO, 0 to 6): 0.00–0.30, negligible; 0.31–0.50, low; 0.51–0.70, moderate; 0.71–0.90, high; 0.91–1.00, very high (Adapted from: Hinkle D et al, 2003). All comparisons were statistically significant ($p < 0.05$). Percent agreement was assessed using kappa coefficients (k): 0%–100% healthy (without warnings) versus less healthy (1 or more warnings). Kappa coefficients were used to categorize agreement as follows: 0.01–0.20, slight; 0.21–0.40, fair; 0.41–0.60, moderate; 0.61–0.80, substantial; 0.81–0.99, near perfect (Viera A et al, 2005).
(DOCX)

**S1 Text. Appendix.** Codebook for photographic methods for measuring packaged food and beverage products in supermarkets and capturing information by fieldworkers.
(DOCX)

**S2 Text. STROBE checklist.** STROBE checklist for the present study.
(DOC)

## Author Contributions

**Conceptualization:** Alejandra Contreras-Manzano, Carlos Cruz-Casarrubias, Claudia Nieto, Simón Barquera.

**Data curation:** Alejandra Contreras-Manzano, Carlos Cruz-Casarrubias, Ana Munguía, Jorge Vargas-Meza.

**Formal analysis:** Alejandra Contreras-Manzano, Carlos Cruz-Casarrubias.

**Funding acquisition:** Simón Barquera.

**Investigation:** Alejandra Contreras-Manzano, Carlos Cruz-Casarrubias, Lizbeth Tolentino-Mayo.

**Methodology:** Alejandra Contreras-Manzano, Carlos Cruz-Casarrubias, Ana Munguía, Alejandra Jáuregui, Claudia Nieto, Lizbeth Tolentino-Mayo.

**Project administration:** Lizbeth Tolentino-Mayo.

**Resources:** Lizbeth Tolentino-Mayo.

**Software:** Alejandra Contreras-Manzano, Jorge Vargas-Meza, Lizbeth Tolentino-Mayo.

**Supervision:** Alejandra Contreras-Manzano, Simón Barquera.

**Validation:** Alejandra Contreras-Manzano, Ana Munguía.

**Writing – original draft:** Alejandra Contreras-Manzano, Carlos Cruz-Casarrubias, Alejandra Jáuregui.

**Writing – review & editing:** Alejandra Contreras-Manzano, Carlos Cruz-Casarrubias, Ana Munguía, Alejandra Jáuregui, Jorge Vargas-Meza, Claudia Nieto, Lizbeth Tolentino-Mayo, Simón Barquera.

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
