## [Editor Report · Decision Letter 0]

9 Jun 2021

Dear Dr Jauregui, 

Thank you for submitting your manuscript entitled "Validation of the Mexican Warning Label Nutrient Profile." for consideration by PLOS Medicine.

Your manuscript has now been evaluated by the PLOS Medicine editorial staff and I am writing to let you know that we would like to send your submission out for external peer review.

Please re-submit your manuscript within two working days, i.e. by Jun 11 2021 11:59PM.

Kind regards,

Beryne Odeny

Associate Editor

PLOS Medicine

---

## [Decision Letter · Decision Letter 1]

8 Feb 2022

Dear Dr. Jauregui,

Thank you very much for submitting your manuscript "Validation of the Mexican Warning Label Nutrient Profile." (PMEDICINE-D-21-02443R1) for consideration at PLOS Medicine. We do apologize for the delay in sending you a response. 

Your paper was discussed with an academic editor with relevant expertise and sent to independent reviewers, including a statistical reviewer. The reviews are appended at the bottom of this email and any accompanying reviewer attachments can be seen via the link below:

[LINK]

In light of these reviews, we will not be able to accept the manuscript for publication in the journal in its current form, but we would like to invite you to submit a revised version that addresses the reviewers' and editors' comments fully. You will appreciate that we cannot make a decision about publication until we have seen the revised manuscript and your response, and we expect to seek re-review by one or more of the reviewers. 

We hope to receive your revised manuscript by Mar 01 2022 11:59PM. Please email us (plosmedicine@plos.org) if you have any questions or concerns.

Please let me know if you have any questions, and we look forward to receiving your revised manuscript. 

Sincerely,

Richard Turner, PhD

Senior editor, PLOS Medicine

rturner@plos.org

Our academic editor commented: "I disagree with the term "validity" in this context. As the authors discuss, there is no gold standard to be considered ...", and we therefore suggest adapting the language used throughout the paper in favour of "evaluation" and "comparison" between models.

Please adapt the title to better match journal style. We suggest: "Evaluation of the Mexican warning label nutrient profile: A population-based cohort study". 

Please restructure the abstract into three subsections (i.e., Background, Methods and findings, and Conclusions). 

Please add a new final sentence to the "Methods and findings" subsection, which should begin "Study limitations include ..." and should quote 2-3 of the study's main limitations. 

After the abstract, please add a new and accessible "Author summary" section in non-identical prose. You may find it helpful to consult one or two recent research papers in PLOS Medicine to get an impression of the preferred style. 

Early in the Methods section (main text), please state whether or not the study had a protocol or prespecified analysis plan, and if so attach the relevant file, and referred to in the text. 

Please correct "strengths" at line 461.

Please avoid assertions such as "the first" at line 464, and where needed add "to our knowledge" or similar. 

Throughout the text, please reformat reference call-outs as follows: "... and calories [1,2]." (noting the absence of spaces within the square brackets). 

Please remove all iterations of "[Internet]" from the reference list.

Please revisit reference 6, which may be lacking a journal name or URL.

Please include a completed checklist for the most appropriate reporting guideline, e.g., STROBE, as an attached file, labelled "S1_STROBE_Checklist" or similar and referred to as such in the Methods section (main text). 

In the checklist, please refer to individual items by section (e.g., "Methods") and paragraph number, not by line or page numbers as these generally change in the event of publication. 

Thank you for supplying the attached data file. However, this may be challenging to access, and we suggest providing Excel files instead, or using an online repository. 

Comments from the reviewers:

*** Reviewer #1: 

Review of Validation of the Mexican Warning Label Nutrient Profile.

The authors compare the agreement and correlation of the Mexican NP system with the Pan American recommendations and those of other Latin American countries using Cohen's kappa and Pearson's correlation. The sample size is large and impressive. The graphs are clear and make understanding of this complex issue easier to grasp. The article is over-all well-written and well-presented. I have one minor and one major recommendation.

Methods: 

Line 174: Probably should be 'We selected a convenience sample'. Some indication as to how the stores were selected should be included, even if it's just "We chose stores within walking distance of the university" or whatever. 

Results: 

No p-values or confidence intervals are reported anywhere in the article. This is somewhat understandable due to the density of the data presentation and the large sample size is likely to produce all significant p-values. However, this is a grave lapse in rigorous scientific reporting and it is essential that either p-values or confidence intervals are reported. STATA should produce both p-values and confidence intervals for both the Agreement and Correlation analyses. At a minimum, if all p-values are less than 0.001 (for example), then this should be stated somewhere in the text and in the table headings where kappa and the correlation are reported. 

*** Reviewer #2: 

[see attachment]

*** Reviewer #3: 

I mostly confine my remarks to statistical aspects of this paper. I would mostly change the emphaiss, and I have some issues with the figures.

If I understand correctly, one method of validation was to see what proportion of the total items were called XXX (e.g. unhealthy) in the different NPs. This can't be a very good method of validation. What if the proportions are similar but the foods are different. e.g. if there were only 5 foods (A, B, C, D, E) and one NP labeled A and B as unhealthy while the other labeled D and E, then they would each label 40% but this would be an indication of total lack of validiity. So, I would drop that from the write up and just go with kappas across the various foods (I am assuming kappa was calculated for each food, so that you had thousands of 2x2 tables to work with).

I'm not going to comment much on the validity of the whole plan --- I am not a nutritionist --- but on p. 9, line 249, it appears that raw or unprocessed food was not labeled. How can this be right? IN that case, pure butter or olive oil or sugar would not get warnings for being high in calories or fat; sugar would not be high in sugar! I'd think it would be better to distinguish ingredients from finished products. After all, no one makes a meal of just butter or just sugar. Also, earlier the authors talk about dairy products being labeled, does this mean they were only talking about dairy products that are processed? This seems unclear.

Back to statistical issues. For the graphs: None of these are good. Stacked bar plots are not a good graphical method. See the work of William Cleveland. I would suggest Cleveland dot plots. Since you are using Stata, you could use the multidot command

Peter Flom

***

[LINK]

---

## [Decision Letter · Decision Letter 2]

12 Mar 2022

Dear Dr. Jauregui,

Thank you very much for re-submitting your manuscript "Evaluation of the Mexican Warning Label Nutrient Profile: A cross-sectional analysis of the Mexican market share in 2016 and 2017." (PMEDICINE-D-21-02443R2) for consideration at PLOS Medicine.

I have discussed the paper with our academic editor and it was also seen again by three reviewers. I am pleased to tell you that, provided the remaining editorial and production issues are fully dealt with, we expect to be able to accept the paper for publication in the journal.

[LINK]

Please let me know if you have any questions in the meantime, and we look forward to receiving the revised manuscript.   

Sincerely,

Richard Turner, PhD

rturner@plos.org

Requests from Editors:

We ask you to adapt the title to better match journal style, and suggest: "Evaluation of the Mexican warning label nutrient profile on food products marketed in Mexico in 2016 and 2017: A cross-sectional analysis".

Please remove the hyphen from "Latin America" at line 31 and any other instances.

Please add "... nutrient profiles (NP)" at line 31 and make that "Mexican NP" in the following line. 

At line 35 we suggest removing "... using the calibration method.", and adding a sentence, say, early in the "Methods and findings" subsection of the abstract to explain briefly what this method involves. 

Please make that "3 implementation phases" at line 39, and use this style throughout the article (although numbers should not be used at the start of sentences). 

At line 39, do you need to state briefly when the implementation phases were/will be?

At line 47, please adapt the text to "... were those implemented in Ecuador ..." or similar. 

At line 53, please make that "... are limited"; and "data are ..." at line 562.

At line 53, please rephrase "... as if they were octagon warnings", which may not be clear to all readers (e.g., you could add a few words to explain what an "octagon warning" is. A brief explanation may be needed later in the ms too. 

At line 53, why do you highlight the Brazil NP when those from three countries appear to have low agreement with the PAHO model? If appropriate, all three countries could be listed here.

Please adapt the author summary so that there are three subsections, each comprising about three points (the third subsection should be titled "What do these findings mean?".

At line 89 (author summary) please make that "We compared ...".

Thank you for adding information about the study protocol. PLOS Medicine's aim is for readers to be able to appreciate to what extent a study's analyses were preplanned, or to the contrary could have been data-driven. Therefore, please add a statement in the Methods section (main text) of the type: "This study did not have a prespecified analysis plan"; or "The study's analyses were preplanned, except for ...", for example.

Please refer to the STROBE checklist in the Methods section (main text), e.g., "The study is reported according to STROBE (see S1_STROBE_Checklist).", and you may wish to add a reference to the primary STROBE publication. 

Does "WL" need to spelt out at line 557?

Please remove the information on funding from the end of the main text. In the event of publication, this information will appear in the article metadata, via entries in the submission form. 

Noting references 33 & 47, please format the authors' names as for other references. 

Comment from Academic Editor:

One minor comment I would suggest would be to include a bit more background in their Introduction and Author summary about the Mexican NP model. As they explained, the PAHO model was created for Latin American countries as reference, however, each country including Mexico have developed their own NP and it's unclear why. Why is the Mexican NP needed then? What is the Mexican NP trying to do that the PAHO is not doing? Both methods are in high agreement and are very similar, so I think the audience would appreciate a bit more information on why the Mexican NP method (and hence its evaluation against PAHO) is needed. Perhaps this should also be briefly mentioned in Discussion.

Comments from Reviewers:

*** Reviewer #1: 

Thank you for your efforts. 

*** Reviewer #2: 

The revised manuscript adequately addressed my concerns from the prior review. I believe it will make a terrific contribution to the literature in global public health and nutrition.

*** Reviewer #3:

One remaining issue is the graphs. I said they were not good, they said they had improved them. But I do not see any changed graphs. 

Did I miss something? Or did the authors fail to update the figures in the text with new figures? 

Peter Flom

***

[LINK]

---

## [Editor Report · Decision Letter 3]

16 Mar 2022

Dear Dr Jauregui, 

On behalf of my colleagues and the Academic Editor, Dr Piernas, I am pleased to inform you that we have agreed to publish your manuscript "Evaluation of the Mexican warning label nutrient profile on food products marketed in Mexico in 2016 and 2017: A cross-sectional analysis." (PMEDICINE-D-21-02443R3) in PLOS Medicine.

Prior to final acceptance, please address the following points:

At line 40, you may wish to adapt the text to "... 3 implementation phases of increasing stringency ..." or similar; and

Please split the single point under "What did the authors do and find?" in the Author Summary into two, and we suggest the first text could consist of the text "Our findings add support to the relevance of using the Mexican ...", with the second point comprising the comment about limitations. 

PRESS

Sincerely, 

Richard Turner, PhD 

rturner@plos.org